# Snow cover dynamics in Andean watersheds of Chile (32.0-39.5°S) during the years 2000 - 2016

**Alejandra Stehr[1,2,] and Mauricio Aguayo[1,2]**

[1]Centre for Environmental Sciences EULA-CHILE, University of Concepción, Concepción, Chile

[2]Faculty of Environmental Sciences, University of Concepción, Concepción, Chile

*Correspondence to:* A. Stehr (astehr@udec.cl)

**Abstract.** Andean watersheds present important snowfall accumulation mainly during the winter, which melts during the spring and part of the summer. The effect of snowmelt on the water balance can be critical to sustain agriculture activities, hydropower generation, urban water supplies and wildlife. In Chile, 25% of the territory between the region of Valparaiso and Araucanía
comprises areas where snow precipitation occurs. As in many other difficult-to-access regions of the world, there is a lack of hydrological data of the Chilean Andes related to discharge, snow courses, and snow depths, which complicates the analysis of important hydrological processes (e.g. water availability). Remote sensing provides a promising opportunity to enhance the assessment and monitoring of the spatial and temporal variability of snow characteristics, ~~like~~ such as the Snow Cover Area (SCA) and Snow Cover Dynamic (SCD). With regards to the foregoing, the objective of the study is to evaluate the
spatiotemporal dynamics of the SCA at five watersheds (Aconcagua, Rapel, Maule, Biobío and Toltén) located in the Chilean Andes, between latitude 32.0ºS and 39.5ºS, and to analyze its relationship with the precipitation regime/pattern and ENSO events. . Those watersheds were chosen because of their importance in terms of their number of inhabitants, and economic activities depending on water resources. The SCA area was obtained from MOD10A2 for the period 2000–2016, and the SCD was analysed through a number of statistical tests to explore observed trends. In order to verify the SCA for trend analysis, a
validation of the MOD10A2 product was done, consisting on the comparison of snow presence predicted by MODIS with ground observations. Results indicate that there is an overall agreement of 81% to 98% between SCA determined from ground observations and MOD10A2, showing that the MODIS snow product can be taken as a feasible remote sensing tool for SCA estimation in South-Central Chile. Regarding SCD, no significant reduction in SCA for the period 2000-2016 was detected, with the exception of the Aconcagua and Rapel watersheds. In addition to that, an important decline in SCA in the five watersheds for
the period of 2012 and 2016 was also evident, which is coincidental with the rainfall deficit for the same years. Findings were compared against ENSO episodes occurred during 2010–2016, detecting that Niña years are coincident with maximum SCA during winter in all watersheds.

# 1    Introduction

Snowmelt-driven watersheds systems are highly sensitive to climate change, because their hydrologic cycle depends on both precipitation and temperature, and because water is already a scarce resource subject to an ever-increasing pressure for its use (Barnett et al. 2005, Vicuña et al. 2011, Meza et al. 2012, Valdés-Pineda et al. 2014). Snowmelt controls the shape of the annual hydrograph, and affects the water balance at monthly and shorter time scales (Verbunt et al., 2003, Cortés et al. 2011). The effect of snowmelt on the water balance can be critical to sustain agriculture activities, hydropower generation, urban water supply and wildlife habitat quality (e.g. Vicuña et al., 2012 and 2013).

The Andean watersheds present an important snowfall accumulation mainly during the austral winter; snow melts during spring and usually also during part of the summer, depending on relative altitude and ambient temperature. At higher elevations, a snowpack stores significant volumes of water, which are released to the surface runoff and groundwater when solar radiation increases.

In particular, 25% of the Chilean territory between the Valparaiso and Araucanía regions is contained in areas where snow precipitation occurs (DGA, 1995). As in many other difficult-to-access regions of the world, the Chilean Andes - unlike western North America or the European Alps- have limited availability in temporal and spatial extent of hydrological data like discharge data, snow courses, and snow depths (Ragettli et al., 2013), which complicates the analysis of important hydrological processes and the validation of water quantity prediction models.

In this regard, remote sensing provides a promising opportunity to enhance the assessment and monitoring of the spatial and temporal variability of different variables involved in the precipitation-runoff processes in areas where data availability for hydrological modelling is scarce (Simic et al., 2004; Boegh et al., 2004; Melesse et al., 2007; Montzka et al., 2008; Milzowa et al., 2009, Er-Raki et al., 2010; Stehr et al., 2009 and 2010).

Satellite-derived SCA from products like NOAA-AVHRR or MODIS can be used to enhance the assessment and monitoring of the spatial and temporal variability of snow characteristics (Lee et al., 2006; Li & Wang, 2010; Marchane et al., 2015; Wang et al., 2015), especially when it is combined with field data and snowpack models (Kuchment et al., 2010).

Specific spectral reflectance of snow (higher reflectance in the visible spectrum, compared to the mid infrared electromagnetic spectrum) allows SCA to be accurately discriminated from snow-free areas using optical remote sensing methods (in the absence of clouds or vegetation canopies). Compared with other remote sensing techniques such as microwave - which can be used to map snow water equivalent (SWE) - optical remote sensing, which is used to map snow areal extent (SAE), has a much higher spatial resolution (Zhou et al., 2005; Zeinivand & De Smedt, 2009).

P ,,a Previous studies have compared MODIS snow maps with ground observations and snow maps produced by The National Operational Hydrologic Remote Sensing Centre, USA (NOHRSC) (Hall et al., 2002; Klein & Barnett, 2003; Tekeli et al., 2005; Aulta et al., 2006). Klein & Barnett (2003) compared MODIS and NOHRSC products with ground observations (SNOTEL measurements), obtaining an overall accuracy of 94% and 76%, respectively. The MOD10A2 snow product is capable of predicting the presence of snow with good precision (over 90%) when the sky is clear (Zhou et al., 2005; Liang et al., 2008a; Wang et al., 2008; Huang et al, 2011, Wenlong et al. 2017). All the previous studies were done in watershed with a not so

complex surface topography as in the Chilean Andes, where we have very steep slopes and a great oceanic effect due to the short distance between the coastal line and the mountains.

Many of the studies on SCA changes and variability were done in the northern hemisphere, where topographic and orographic conditions are different from the ones we have in Chile. Research studies generally show negative trends in snow cover extent and snow water equivalent across both North America and Eurasia.

In Europe, Krajci et al. (2016) analysed the main Slovak watersheds; for the 2001–2006 period, they obtained an increase of mean SCA; however a significantly lower SCA is observed in the next 2007–2012 period. Their results indicate that there is no significant change in the mean watershed SCA in the period 2001–2014. Dietz et al. (2012), in their study of snow cover characteristics in Europe, found some abnormal events when comparing the mean conditions with single snow cover seasons. For the season 2005/2006 in particular, an increased snow cover with a later snow cover melt was detected. Marchane et al. (2015) studied the SCD over the Moroccan Atlas mountain range from 2000 until 2013, concluding that SCA has a strong inter-annual variability and that there is no statistical evidence of trend in that period.

In Asia, two important regions have been studied: the Tibetan Plateau and the Himalayan region. In the case of the Tibetan Plateau, Zhang et al. (2012) have studied SCD of four lake watersheds for the period 2001–2010; results indicate that spatial distribution and patterns of snow-covered days are very stable from year to year, and that there is no trend of snow cover change for each watershed. For the same period, Tang et al. (2013) found a high inter-annual variability of SCA, with parts of the studied area showing a declining trend in SCA, and other parts showing increasing trends in SCA. Wang et al. (2015) evaluated trends in SCA, showing a decrease of snow-covered areas from 2003 until 2010. For the Himalayan region, Maskey et al. (2011) studied the trends and variability of snow cover changes above the 3000 m during the 2000-2008 period, showing a decreasing trend of snow cover during January and increasing trends during March. Snehmani et al. (2016) studied the 2001-2012 winter period (November–April) in order to obtain clear negative snow cover trends in the basin. Azmat et al. (2017) indicated that the watershed shows a consistent or slightly decreasing trend of snow cover, particularly over the high-altitude parts of the watershed during 2000–2009, and in the 14-year analysis (2000–2013), a slight expansion in the snow-covered area was observed in the whole basin. Gurung et al. (2017), based on the analysis SCA data between 2003-2012, also obtained a decline of SCA, having a statistically significant negative correlation between SCA and temperature, which indicates that this trend is partly a result of increasing temperatures.

In North America, Pederson et al. (2013) found that, since 1980, in the northern and southern Rocky Mountains the April 1 snow water equivalent (SWE) has been changing synchronously, and generally declining associated with spring time warming. Fassnacht & Hultstrand (2015) examined trends in snow depth and SWE from three long-term snow course stations in northern Colorado from 1936 to 2014. They found negative trends at two of the stations for all the period and, at the third station, a positive trend was found for the first half of the record and a decrease over the second half. Harpold et al. (2012) analyzed SNOTEL SWE for the central and southern Rocky Mountains for the period 1984-2009; they found widespread decreases in maximum SWE and duration of snow cover. After a review of various studies in North America, Kunkel et al. (2016) concluded that all of them are consistent in indicating a decrease in snow on the ground with some of the most extreme low values occurring in the last 10–15 years.

Studies of snowpack variation done in the Central Andes of Chile and Argentina show that the average regional maximum value of the SWE series displays a positive (though non-significant) trend with marked interannual variability ranging from 6% to 257% of the 1966 –2004 mean (Masiokas et al. 2006). Results from Cornwell et al. (2016), who made a SWE reconstruction between 2001 and 2014, indicate that the years 2002 and 2005 stand out for displaying large positive anomalies throughout the entire model domain. The northern part shows above-average accumulation in only 3 out of the 14 simulated years (2002, 2005 and 2007), whereas the other part of the study area shows above-average accumulation for 6 years (2001, 2002, 2005, 2006, 2008 and 2009). In particular, the years 2007 and 2009 show a bimodal spatial structure, with excess accumulation (deficit) in the northern (southern) area and the inverse pattern in the latter year.

From the literature review, it is evident that SCD is site-specific and exhibits high variability among the different sites around the world. The Andes mountain range (on the Chilean side) has a great oceanic effect due to the short distance between the coastal line and the mountains, unlike the northern hemisphere which has a big continental influence. This makes the isotherm 0 higher than in the northern hemisphere; in addition, the topographic features characterized by mean heights above 3000 m, with very steep slopes that produce a huge orographic effect that forces the rise of the western winds and condensation of moisture, strongly affect the regime of precipitation and temperatures. This causes small-scale spatial variations in weather, which are sometimes difficult to identify in satellite imagery. All the previously mentioned aspects make it necessary to do a validation of MOD10A2, before using it for other analyses. Indeed, one of the main sources of error in the classification of satellite images is the interpretation of topographic features especially in areas with rugged slopes. The irregular topography produces shading and lighting effects that change the radiometric response of the surface, which depends on the local slope and its orientation (Riaño et al., 2003). As the climate-controlling hydrologic processes in the Andes are influenced by El Niño events (Escobar y Aceituno 1993, Ayala et al. 2014), as well as warm winter storms (Garreaud 2013), assessment of SCD is of special interest in this region. Pioneer studies have been presented in recent years for the Mataquito river watershed (De María et al. 2013, Vicuña et al. 2013).TheThe aim of the study is to evaluate the spatiotemporal dynamics of the SCA in the South Central Chilean Andes, and to analyze its relationship with the precipitation regime/pattern and ENSO events. .  To prove this, we selected five watersheds located in the Chilean Andes, between latitude 32.0ºS and 39.5ºS. We investigated trends and variability of snow cover changes at different temporal scales (seasonal and annual) and we used Moderate Resolution Imaging Spectroradiometer (MODIS) snow cover products (Hall et al. 2007) from 2000 to 2017. In-situ precipitation measurements from the National General Water Directorate (DGA) were used.

## 2    Materials and methods

### 2.1    Study Sites

The study site includes five watersheds located in central and southern Chile: Aconcagua, Rapel, Maule, Biobío and Toltén (figure 1). The watersheds were chosen considering their population and dependence on water-resourced economic activities.

The Aconcagua watershed is located in the Valparaiso region, between the parallels 32°14′–33°09′S and 69º59′–71º33′W, with an area of 7340 km$^2$ and a maximum elevation of 5843 m a.m.s.l. Approximately 40% of the watershed lies above the snowline (average altitude above where snow can be found in winter), which is located at 2100 m (Garreaud, 1992). The climate in the

watershed is temperate Mediterranean with a long dry season of 7 to 8 months, and a wet season of approximately 4 months (May–August) during which more than 80% of the precipitation occurs. The average annual precipitation is 529 mm. Principal economic activities at the watershed are agriculture, mining and industries. It has a population of around 600,000 inhabitants (4% of the Chilean population).

The Rapel watershed is located in the General Libertador Bernardo O´Higgins region between the parallels 33°52′–35°00′S and 70º00′–71º53′W, having an area of 13695 km$^2$ and a maximum elevation of 5138 m a.m.s.l.. Approximately 30% of the watershed lies above the snowline which is located at 1500 m (Peña & Vidal, 1993). The climate in the watershed is temperate Mediterranean, with a dry season of 4 to 5 months (November–March), and a wet season of approximately 4 months (May–August) during which more than 75% of the precipitation occurs. The average annual precipitation is 960 mm. The main

economic activities at the watershed are agriculture and mining. It has a population of around 570,000 inhabitants (3.8 % of the Chilean population).

The Maule watershed is located in the Maule region between the parallels 35°05′–36°35′S and 70º18′–72º42′W, having an area of 20295 km$^2$ and a maximum elevation of 3931 m a.m.s.l.. Approximately 32% of the watershed lies above the snowline, which is located at 1150 m (Peña y Vidal, 1993). The climate in the watershed is temperate Mediterranean, with a 6-month dry season

(November–April), and a wet season of approximately 4 months (May–August) during which more than 75% of the precipitation occurs. The average annual precipitation is 1471 mm. The main economic activity at the watershed is agriculture. It has a population of around 410,000 inhabitants (2.7 % of the Chilean population).

The Biobío watershed is located in the Biobio region, between the parallels 36°45′–38°49′S and 71º00′–73º20′W, having an area of 24264 km$^2$ and a maximum elevation of 3487 m a.m.s.l.. Approximately 41% of the watershed lies above the snowline which

is located at 850 m (Peña y Vidal, 1993). The climate in the watershed is Mediterranean, with a 5-month dry season (November–March), and a wet season of approximately 4 months (May–August), during which more than 55% of the precipitation occurs. The average annual precipitation is 1891 mm. Principal economic activities related to water resources at the watershed are agriculture, forestry and industries. It has a population of around 630,000 inhabitants (4.2 % of Chilean population).

The Toltén watershed is located in the Araucanía region, between the parallels 38°32′–39°38′S and 71º21′–73º16′W, having an

area of 8398 km$^2$ and a maximum elevation of 3710 m a.m.s.l.. Approximately 37% of the watershed lies above the snowline which is located at 750 m (Peña & Vidal, 1993). The climate in the watershed is temperate rainforest with Mediterranean influence, characterized by precipitation throughout the year but having less rain during the summer months than in the winter ones. The average annual precipitation is 2870 mm. Main economic activities related to water resources at the watershed are tourism, agriculture and forestry. It has a population of around 170,000 inhabitants (1.1 % of the Chilean population).

**2.2    Ground observations of SCA**

For validation of MOD10A2 ground observations were performed, including continuous monitoring of snow depth with meteorological stations, snow courses, and one-day observations of snow presence and depth at some mountain trails. Figure 2a shows the location of measuring sites.

### 2.2.1 Continuous monitoring of snow depth

In order to perform a continuous measurement of snow depth, three meteorological stations were installed in the upper part of the Biobío watershed. The stations are Parque Tolhuaca, located at 900 m.a.s.l., Termas Malleco at 1190 m.a.s.l., and Laguna Verde at 1410 m.a.s.l.. Also, snow depth data from three DGA meteorological stations (Portillo 3005 m.a.s.l., Laguna Negra 2709 m.a.s.l. and Volcan Chillan 1964 m.a.s.l.) were used. Figure 2b shows the location of the meteorological stations.

Snow depth was measured using acoustic snow depth sensors (Campbell Scientific, SR50A) with a frequency of 15 min. Corrections for variation of the speed of sound in air were made considering the air temperature measured at the same time intervals as the snow depth, using the temperature and relative humidity probe (Vaisala, HMP60). The data were collected from April 2010 to December 2011 at the Termas Malleco station, and from July 2011 to December 2011 at the Parque Tolhuaca and Laguna Verde stations. In the case of the DGA station, available data from April 2013 to December 2015 was used. Snow data were grouped considering the average snow depth over 8 days (same 8-day period of MOD10A2) and then reclassified as snow (1) if the average snow depth was > 0 cm and no-snow (0) for snow depth = 0 cm.

### 2.2.2 One-day observation points

To cover a large spatial domain and to achieve a better spatial and temporal representation of SCA, a total of 124 different single measurements of snow depth were conducted during field campaigns at different mountain trails. Field measurements were taken since the end of June until the beginning of October 2011. The location of each observation point was recorded with a GPS and snow depth was measured with a Black Diamond QuickDraw Tour Probe 190. Figure 2c shows the location of the observation points and the date and number of observations for each day.

### 2.2.3 Snow courses

During six days of the 2011 winter season, eleven snow courses were conducted in the upper Malleco watershed. Figure 2d shows the location of snow courses. Each route was recorded with a GPS. Snow depth and density were measured with a Black Diamond QuickDraw Tour Probe 190 and Snow Sampling Tubes (3600 Federal Snow Tubes, Standard-Metric), respectively. Table 1 shows the dates and repetitions for each snow route performed.

### 2.3 MODIS snow cover products

The Moderate Resolution Imaging Spectroradiometer (MODIS) is on the Earth Observing System, which employs a cross-track scan mirror and a set of individual detector elements to provide imagery of the Earth surface and clouds in 36 discrete and narrow spectral bands ranging in wavelength from 0.405 to 14.385 μm. It provides medium-to-coarse resolution imagery with a high temporal repeat cycle (1-2 days). The main purpose of the MODIS is to facilitate the study of global vegetation and land cover, vegetation properties, global land surface changes, surface albedo, surface temperature as well as snow and ice cover, on a daily or nearly daily basis. The MODIS snow cover products are one of the many geophysical standard products derived from MODIS data. The MODIS snow cover products are provided on a daily basis (MOD10A1) and as 8-day composites (MOD10A2), both at 500 m resolution over the Earth's land surfaces. MOD10A1 consists of 1200 km by 1200 km tiles of 500 m resolution data gridded in a sinusoidal map projection and MOD10A2 is a composite of MOD10A1 especially produced to show

maximum snow extent. For this study, the MOD10A2v005 (Hall et al. 2016) was used. Classification of SCA using MODIS collected data was done based on the Normalized Difference Snow Index (NDSI) (Hall et al., 1995, Klein et al., 1998, Riggs et al., 2006). Images were reprojected to WGS84 UTM 19S using the MODIS Reprojection Tool (MRT).

MODIS estimates of SCA were validated for the period between April 2010 and December 2011 by comparing them with ground observations. Eighty snow maps from MOD10A2 were analysed. Only cloud-free observations for each cell of the map grid were used.

### 2.4    Validation of MOD10A2

To validate the correspondence between the image classification and ground observations, a confusion matrix was used (Congalton & Mead, 1983; Congalton, 1986, Congalton, 1991; Foody, 2002). The confusion matrix is a simple cross tabulation of classified data versus observed ones, providing a base for accuracy assessment (Campbell, 1996; Canters, 1997). Figure 3 present the confusion matrix and how the different indexes were calculated.

### 2.5    Assessment of SCD

SCA variation in the five selected watersheds was evaluated using MOD10A2 for the period covering the years 2000–2016 using a total of 774 images. All images were processed in ArcGis, reprojecting them to WGS84 19S and cut according to the desired area, in this case for the Aconcagua, Rapel, Maule, Biobío and Toltén watersheds. SCA and clouds were quantified for each available image, and annual and seasonal averages of snow cover were calculated for each watershed.

SCD was analysed through a number of statistical tests to explore observed trends in SCA during the study period. "Nonparametric" statistical test were chosen because they are robuster than the "parametric" test. Mann-Kendall test was applied to determine the existence of monotonic trends. Sen's method (Gilbert, 1987) was applied to determine the rate of observed changes.

An analysis of correlation between SCA and mean annual precipitation for the upper part of the watershed  was done. Considering the spatial location of SCA, only precipitation stations located over the 700 m.a.s.l. were considered. Availability of daily precipitation was evaluated for each year at each station; only years with more than 80% of data were used in order to obtain the mean annual precipitation. The precipitation for the watershed was obtained considering the arithmetic mean between available stations. Table 2 shows the stations that were used and the availability of data.

### 3    Results

The product MOD10A2 was validated for determination of SCA in the watersheds under study and the SCD was analysed.

### 3.1 Validation of MOD10A2 through ground observations

The composite images MOD10A2 were compared with ground observations. For the study period 23, 75, 26, 119, 123 and 32 images were available for comparison with observations at Parque Tolhuaca, Termas Malleco, Laguna Verde, Portillo, Laguna Negra and Volcan Chillan stations, respectively. Clouds were present in 2 (9%), 1 (1%), 1 (4%), 3 (3%), 3 (2%) and 2 (6%) images for each aforementioned station, respectively. Table 3 present the confusion matrix and the indexes of agreement. An overall accuracy of 86%, 81%, 88%, 92 %, 83% and 97% was observed at Parque Tolhuaca, Termas Malleco, Laguna Verde, Portillo, Laguna Negra and Volcan Chillan stations, respectively, which reached the target of 85% (Thomlinson et al., 1999).

A total of 117 images corresponding to the study period were available for comparison with one-day observations, with none of them classified as covered by 'clouds'. All ground observations were done over areas with snow presence. When comparing it to MOD10A2 images the agreement was of 97%. MODIS that did not coincide with ground observations occurred during the beginning of the snowfall season or at the end of the melting period. In both cases, snow patches were observed in the field covering areas at the subpixel scale of the MOD10A2 image.

For comparison with snow-courses, a total of 282 images were available during the study period, all of them without clouds. Ground observations were done on areas covered by snow only. The overall accuracy of MOD10A2 for predicting SCA was 98%.

Results indicate that MOD10A2 has a satisfactory agreement with ground observations, and therefore the 8-day composite images are suitable for analysis of SCD in the Andean watersheds.

### 3.2 Assessment of SCD

Figure 4 shows the SCD in Aconcagua, Rapel, Maule, Biobío and Toltén watersheds for the period 2000-2016. All watersheds show the same SCD with more SCA in winter than the other seasons. It can be appreciated that in the Maule, Biobío and Toltén watersheds the maximum SCA in 2016 is considerably lower than in the five previous years, and one lowest of the whole study period.

Figure 5 shows mean SCA for each season and annual precipitation in the Aconcagua, Rapel, Maule, Biobío and Toltén watersheds for the period 2000-2016. Northern watersheds, i.e. Aconcagua and Rapel, present a higher SCA percentage. It can be appreciated that not necessarily years with more precipitation have more SCA; instead, years with lower precipitation have a higher SCA. Figure 6display the relation between SCA and annual precipitation for the different watersheds; it is clear that only at Aconcagua there is a good adjustment to the regression line ($R^2 = 0.7$) with a positive slope. In Rapel and Maule we can see a positive relation but with $R^2$ smaller than 0.5, which indicates a deficient adjustment. At Biobío we can graphically see a negative slope, i.e. when there is more precipitation there is less SCA, but statistically we have no adequate adjustment.

Figure 7 shows the variability of spatial distribution of maximum yearly SCA for the five watersheds. Years with maximum and minimum SCA of yearly maximum SCA are not coincident between watersheds. A considerable spatial variability of maximum SCA can be appreciated. There are no trends in the data.

Figure 8 shows annual and seasonal trends of SCA at the Aconcagua, Rapel, Maule, Biobío and Toltén watersheds. Trend analysis to the SCA series was performed with the nonparametric Mann-Kendall test. Decreasing trends in annual mean SCA are observed (p-value < 0.01) for the Aconcagua and Rapel watersheds, with a decreasing slope of 30 $km^2$/year (Figure 8a,b). No significant annual trend was observed for the other three watersheds. In autumn (Figure 8a), only the Aconcagua watershed shows a decreasing trend of mean SCA variation at a level of significance <= 0.05, with a decreasing slope of 54 $km^2$/year. There is no significant variation in SCA in the winter (Figure 8). In spring (Figure 8b,c) the Rapel and Maule watersheds have a decreasing trend at a level of significance <= 0.05, with a decreasing slope of 38 $km^2$/year and 79 $km^2$/year, respectively. In summer (Figure 8c), the Rapel watershed has a decreasing trend at a level of significance <= 0.05, with a slope of 24 $km^2$/year; the Aconcagua and Maule watersheds show some decreasing trends with level of significance <= 0.1 and slopes of 10 $km^2$/year and 11 $km^2$/year (Figure 8a,c). It is important to remark that the Rapel watershed has large glacier areas. All above mentioned results are coincident with the outcomes given by the Pettitt homogeneity test, which shows that time series are not homogenous between two given times (p-value < 0.01). The change in the average in most of the cases is observed between 2008 y 2011.

Considering snow accumulation and melt seasons, data were grouped in two periods, i.e. autumn-winter (accumulation) and spring–summer (melt). For these two periods, a trend analysis was also done. Results indicated that there is a negative trend in average SCA during both seasons; the Aconcagua watershed has a decreasing slope (p-value < 0.01) of 29 $km^2$/year during the snow accumulation season and the Maule watershed has a decreasing slope (p-value < 0.05) of 26 $km^2$/year at the melt season. In the case of yearly maximum SCA, we found a significant positive trend for the Biobío watershed (p-value < 0.01), the other watersheds shown a nonsignificant linear trend.

## 4    Discussion

In general terms, we can say that MOD10A2 has a good agreement with ground observations with always over a 80% of precision when the sky is clear. Our results are a bit lower than the 90% obtained in studies of the northern hemisphere (Zhou et al., 2005; Liang et al., 2008a; Wang et al., 2008; Huang et al, 2011, Zheng et al. 2017), this difference can be attributed to the complex topography of the Chilean Andes Greater disagreements between MOD10A2 and ground observations were found at Termas Malleco and Laguna Negra stations (. These stations are located close to the snowline where most of the time the soil is not covered by snow or is covered by snow for small periods of time; in this case, the presence of small snow patches is expected to be highly variable in time, it can even change at the subdaily scale and therefore it could reduce the capacity of images to capture snow coverage in the proximity of the snowline. Regarding SCA, we can see in the results presented on Figures 4 and 5 that SCA has a similar behaviour in the different watersheds during the study period, i.e. all watersheds follow the same snow accumulation and melt dynamics and the maximum and minimum % of SCA are also coincident in time. SCA has a strong inter-annual variability, what is in agreement with the results obtained by other studies (Masiokas et al. 2006, Marchane et al. 2015, Tang et al. 2013, Zhang et al. 2012).Comparing the result with the obtained by Karici et al (2016) in Slovak watersheds we can appreciate similar results for the Aconcagua, Rapel and Maule watersheds and opposite ones for Biobío and Toltén, i.e. the 2001–2006 period an decrease of mean SCA whit respect to the mean of the whole period; and more SCA for the period 2007–2012.

Comparing our results of SCA with those of SWE obtained by Cornwell et al. (2016), we can appreciate that they are in agreement only with respect that the years 2002 and 2005 display the largest SCA of the entire study period. The other results are in dissension, as we have more years with above mean SCA during the beginning of the study period, and below throughout the end of the period (2012 – 2016).

In the case of trend analysis only the watershed located in the northern part of the study area have a significant decreasing trend during accumulation an melt season, what is in agreement with results from different studies in Asia and North America (Tang et al. 2013, Mackey et al. 2011, Gurung et al. 2017, Fassnacht & Hultstrand 2015, Kunkel et al. 2016). During winter time we could see any significant trend what consistent with results from Dietz et al. (2012).

The SCA magnitude for each watershed was contrasted with historical data from El Niño/ La Niña Southern Oscillation (ENSO)
episodes from 2000–2016[1]. Normal or neutral years correspond to historical conditions, as Niño and Niña years correspond to wet and dry years respectively, in the case of central and southern Chile. During 2000 and 2016, five episodes of El Niño (2002, 2004-2005, 2006 – 2007, 2009 and 2015-2016) and three episodes of la Niña (1999 – 2000, 2007 – 2008 and 2010 – 2011) occurred. Niño and Niña episodes were defined using the Oceanic Niño Index (ONI)

Niña years are coincident with maximum SCA in wintertime in all watersheds, with similar amounts of SCA between different
Niña episodes. The 2007 Niña event is coincident with the highest % of SCA during winter for the study period (2000–2016) in all watersheds. Winter SCA for Niña and Niño episodes are all above or just at the mean SCA for the 14 years under study. Years under the mean of SCA area are all not coincident with Niña and Niño episodes, i.e. normal years are all under the mean SCA value. There is also no linear relation between the amount of precipitation and SCA, in exception of the Aconcagua watershed (Figure 6). This study only analyzed snow coverage and not snow depth, i.e. volume of snow, which implies that a
bigger SCA is not necessarily related with more water stored as snow. Considering the existing data availability in time and space of the DGA regarding snow depth, it was not possible to carry out an analysis of changes in snow depth in this study. During the course of the last five years, an important decline in SCA was perceived in the entire watershed, with exception of the Biobío watershed, where no difference can be seen until the last year (2016). This decline is bigger at the Toltén watershed, where the average for the last five years is 50% of the mean for the entire period. In the analysis of the precipitation data we can
observe that in all watershed there was a rainfall deficit during 2012-2016; Aconcagua 26%, Rapel 29%, Maule 24%, Biobío 19%, Toltén 27 % (DGA, 2012, DGA, 2013, DGA, 2014, DGA, 2015, DGA 2016, DMC, 2013, DMC, 2014).

Masiokas et al. (2006) studied snowpack variation between 1951- 2005 in Chile between latitude 30ºS and 37ºS. They found that snow accumulation is positively related to El Niño but they could not find a clear relationship with La Niña. The correlations that they found between snowfall and annual amount of rainfall in central Chile fit well with the known positive correlation of
precipitation with El Niño during the Southern Hemisphere winter (June–August) (Waylen and Caviedes 1990, Montecinos and Aceituno, 2003). It came to their attention that, in contrast to the decreasing trend in snowpack observed across western North America, in the region they studied the average regional maximum value of snow water equivalent shows a positive to nonsignificant trend. In our case, we have only analyzed SCA obtaining a decreasing trend in the watersheds located in the

---

[1] http://www.cpc.noaa.gov/products/analysis_monitoring/ensostuff/ensoyears.shtml

northern part (32.0ºS - 36ºS) and, to the south, a nonsignificant trend (35ºS - 39.5ºS). These results are in agreement with the ones obtained by Falvey and Garreaud (2009), who noticed that, in central and northern Chile (17°–37°S), in situ temperature observations confirm warming in the central valley and western Andes (+0.25°C/decade). In southern Chile (38°S–48°S) temperature trends over land are weak (insignificant at 90% confidence).

In the watersheds located southern than 35ºS we observed no relation between SCA and annual precipitation, which is in agreement with the results of Cortes et al. 2011, who found that all watersheds they studied that were located north of 35ºS had a high and significant correlation to ONI, and that precipitation during el Niño and la Niña episodes seems to be the most important factor controlling CT timing, but with not influence in the southern ones.

**5      Conclusions**

The first validation of MODIS snow product MOD10A2 for estimation of snow covered areas (SCA) via remote sensing in watersheds located in the Southern Hemisphere was presented. Ground observations of SCA were conducted during the years 2010 and 2011 at six study sites including 6 meteorological stations, 124 one-day single-observation points, and 11 snow courses. The SCA was determined for 636 days from MODIS snow products and compared with the SCA measured in situ. The SCA estimated from MOD10A2 presented an overall agreement from 81 to 98% with SCA determined from ground

observations, showing that the MODIS snow product can be taken as a feasible remote sensing tool for SCA estimation in South-Central Chile.

On the other side, we have analyzed SCA trend for the period 2000-2016 in five of the most important watersheds of South-Central Chile in the context of water use and inhabitants, covering a longitudinal gradient from 32°14′S to 39°38′S, which implies different climates regarding precipitation, drier in the northern part than in the southern part. Results indicate that all

watersheds have the same SCD with more snow during winter time, which decreases during spring and summer. Furthermore, a significant negative trend in annual mean SCA is observed for the Aconcagua and Rapel watersheds, which can have implication on water availability for summertime. In general, we can see that there is no significant reduction in SCA for 2000 -2016, with the exception of the Aconcagua watershed. From the data we can appreciate an important decline in SCA for the period of 2012 and 2016, which is coextensive with the rainfall deficit that occurred during the same years.

Results were compared with the ENSO episode during 2000–2016. From the latter comparison we can conclude that in the previously mentioned period, Niña years are coincident with maximum SCA at winter time in all watersheds, with similar amounts of SCA between different Niña episodes.

In summary, the results presented in this work are highly relevant and can be used as one feasible approximation to obtain SCA, particularly in the Chilean Andes because of the lack of hydrological data such as discharge data, snow courses, snow depths,

etc. These can become an outstanding tool to improve the analysis of important hydrological processes and the validation of water quantity prediction models.

**Acknowledgements**

The present research was conducted in the framework of the FONDECYT 11100119 project.

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

**Table 1: Dates and repetition times for each snow route during winter 2011**

| Snow route | Dates | Nº of times route was repeated |
|---|---|---|
| 1 | 30-June | 1 |
| 2 | 30-June | 1 |
| 3 | 30-June, 19-July, 31-August, 14-September | 4 |
| 4 | 1-July, 19-July, 31-August, 15-September | 4 |
| 5 | 1-July | 1 |
| 6 | 1-July | 1 |
| 7 | 1-July | 1 |
| 8 | 31-August, 15-September | 2 |
| 9 | 1-September, 15-September | 2 |
| 10 | 31-August, 14-September | 2 |
| 11 | 31-August, 14-September | 2 |

**Table 2: Precipitation stations and availability of data**

| Watershed | Station | Elevation m a.s.l | Number of years with more than 80% of data | % of Availability |
|---|---|---|---|---|
| Aconcagua | Resguardo Los Patos | 1220 | 16 | 98% |
| | Rio Putaendo En Resguardo Los Patos | 1218 | 11 | 81% |
| | Jahuel | 1020 | 15 | 97% |
| | Los Andes | 820 | 16 | 100% |
| | Rio Aconcagua En Chacabuquito | 950 | 16 | 100% |
| | Vilcuya | 1100 | 16 | 100% |
| | Riecillos | 1290 | 16 | 100% |
| | Las Chilcas | 850 | 16 | 98% |
| Rapel | Rio Pangal En Pangal | 1500 | 13 | 86% |
| | Rio Cachapoal 5 Km. Aguas Abajo Junta Cortaderal | 1127 | 12 | 88% |
| | Central Las Nieves | 700 | 5 | 35% |
| | La Rufina | 743 | 16 | 99% |
| Maule | Fundo El Radal | 685 | 16 | 99% |
| | Vilches Alto | 1058 | 16 | 100% |
| | Hornillo | 810 | 16 | 99% |
| | Rio Melado En El Salto | 730 | 12 | 78% |
| Biobío | Embalse Ralco | 742 | 5 | 35% |
| | Rio Bio-Bio En Llanquen | 767 | 11 | 74% |
| | Laguna Malleco | 894 | 14 | 96% |
| | Liucura | 1043 | 16 | 98% |
| Toltén | Lago Tinquilco | 850 | 16 | 100% |
| | Puesco (Aduana) | 620 | 14 | 94% |

**Table 3. Confusion matrix and indexes of agreement for Parque Tolhuaca, Termas Malleco and Laguna Verde Stations**

| | | Ground observation | | | | | | | | |
|---|---|---|---|---|---|---|---|---|---|---|
| | | Parque Tolhuaca Station | | | Termas Malleco Station | | | Laguna Verde Station | | |
| | | snow | no-snow | User´s accuracy | snow | no-snow | User´s accuracy | snow | no-snow | User´s accuracy |
| MOD10A2 | Snow | 6 | 2 | 0.75 | 22 | 13 | 0.63 | 16 | 1 | 0.94 |
| | no-snow | 1 | 12 | 0.92 | 1 | 38 | 0.97 | 2 | 6 | 0.75 |
| Producer´s accuracy | | 0.86 | 0.86 | | 0.96 | 0.75 | | 0.89 | 0.86 | |
| Overall accuracy | | 0.86 | | | 0.81 | | | 0.88 | | |

| | | Ground observation | | | | | | | | |
|---|---|---|---|---|---|---|---|---|---|---|
| | | Portillo Station | | | Laguna Negra Station | | | Volcan Chillan Station | | |
| | | snow | no-snow | User´s accuracy | snow | no-snow | User´s accuracy | snow | no-snow | User´s accuracy |
| MOD10A2 | Snow | 43 | 7 | 0.86 | 45 | 0 | 1.00 | 11 | 1 | 0.92 |
| | no-snow | 2 | 64 | 0.97 | 21 | 54 | 0.72 | 0 | 18 | 1.00 |
| Producer´s accuracy | | 0.96 | 0.90 | | 0.68 | 1.00 | | 1.00 | 0.95 | |
| Overall accuracy | | 0.92 | | | 0.93 | | | 0.97 | | |

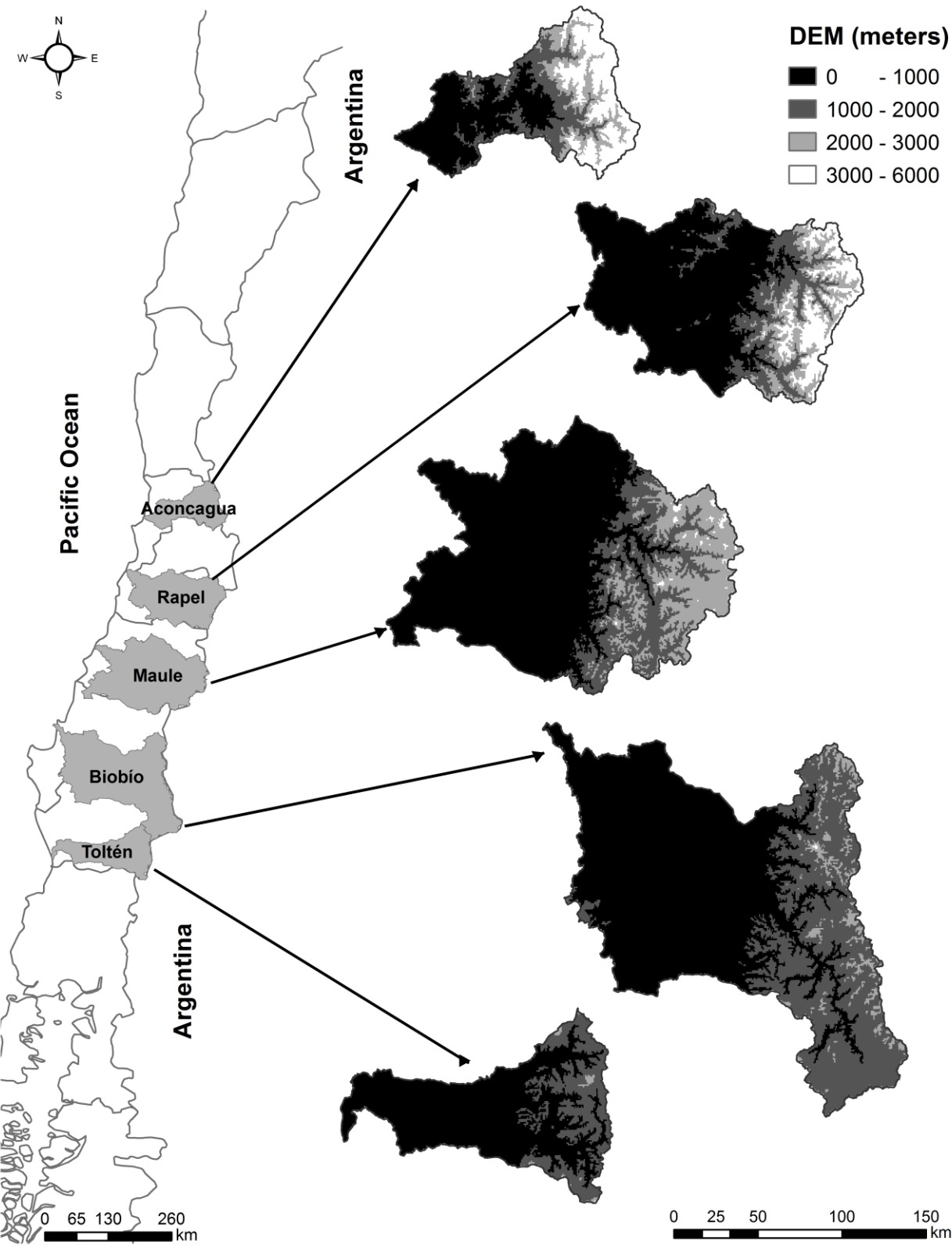

2 **Figure 1: Study sites**

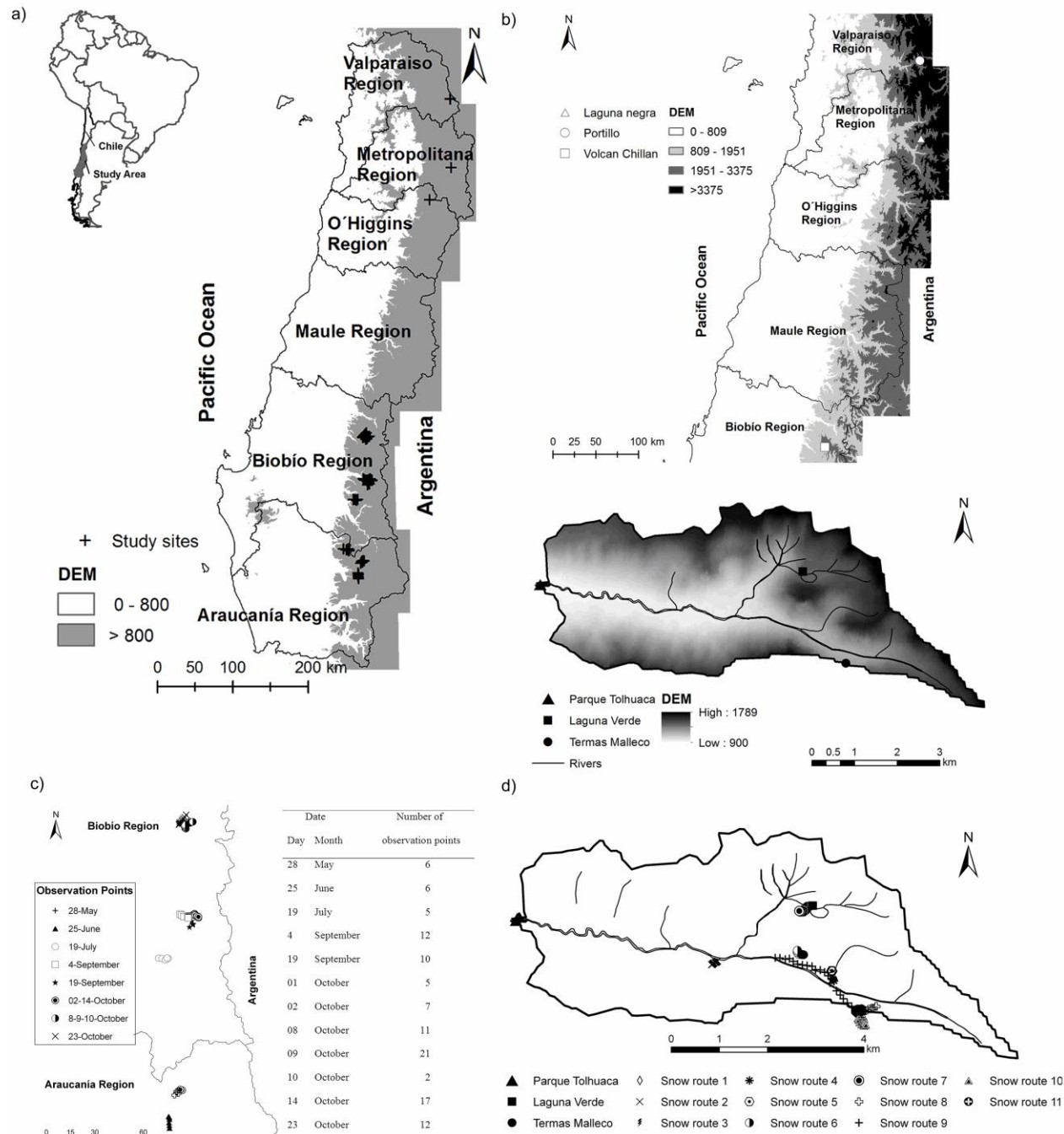

2 **Figure 2: a) Study sites for MOD10A2 validation. b) Location of the meteorological stations for continuous monitoring of**

3 **snow depth. c) Location and date of one-day observation points. d) Location of the snow courses.**

| k,k | A | B | C | ... | q | $\sum$ |
|-----|-----|-----|-----|-----|-----|-----|
| A | $n_{AA}$ | $n_{AB}$ | $n_{AC}$ | ... | $n_{Aq}$ | $n_{A+}$ |
| B | $n_{BA}$ | $n_{BB}$ | $n_{BC}$ | ... | $n_{Bq}$ | $n_{B+}$ |
| C | $n_{CA}$ | $n_{CB}$ | $n_{CC}$ | ... | $n_{Cq}$ | $n_{C+}$ |
| $\vdots$ | $\vdots$ | $\vdots$ | $\vdots$ | ... | $\vdots$ | $\vdots$ |
| q | $n_{qA}$ | $n_{qB}$ | $n_{qC}$ | ... | $n_{qq}$ | $n_{q+}$ |
| $\sum$ | $n_{+A}$ | $n_{+B}$ | $n_{+C}$ | ... | $n_{+q}$ | $n$ |

$$\text{Percentage correct} = \frac{\sum_{k=1}^{q} n_{kk}}{n} \times 100$$

$$\text{User's accuracy} = \frac{n_{kk}}{n_{k+}} \times 100$$

$$\text{Producer's accuracy} = \frac{n_{kk}}{n_{+k}} \times 100$$

2    **Figure 3: Confusion matrix and percentage correct (overall accuracy), user accuracy and producer accuracy.**

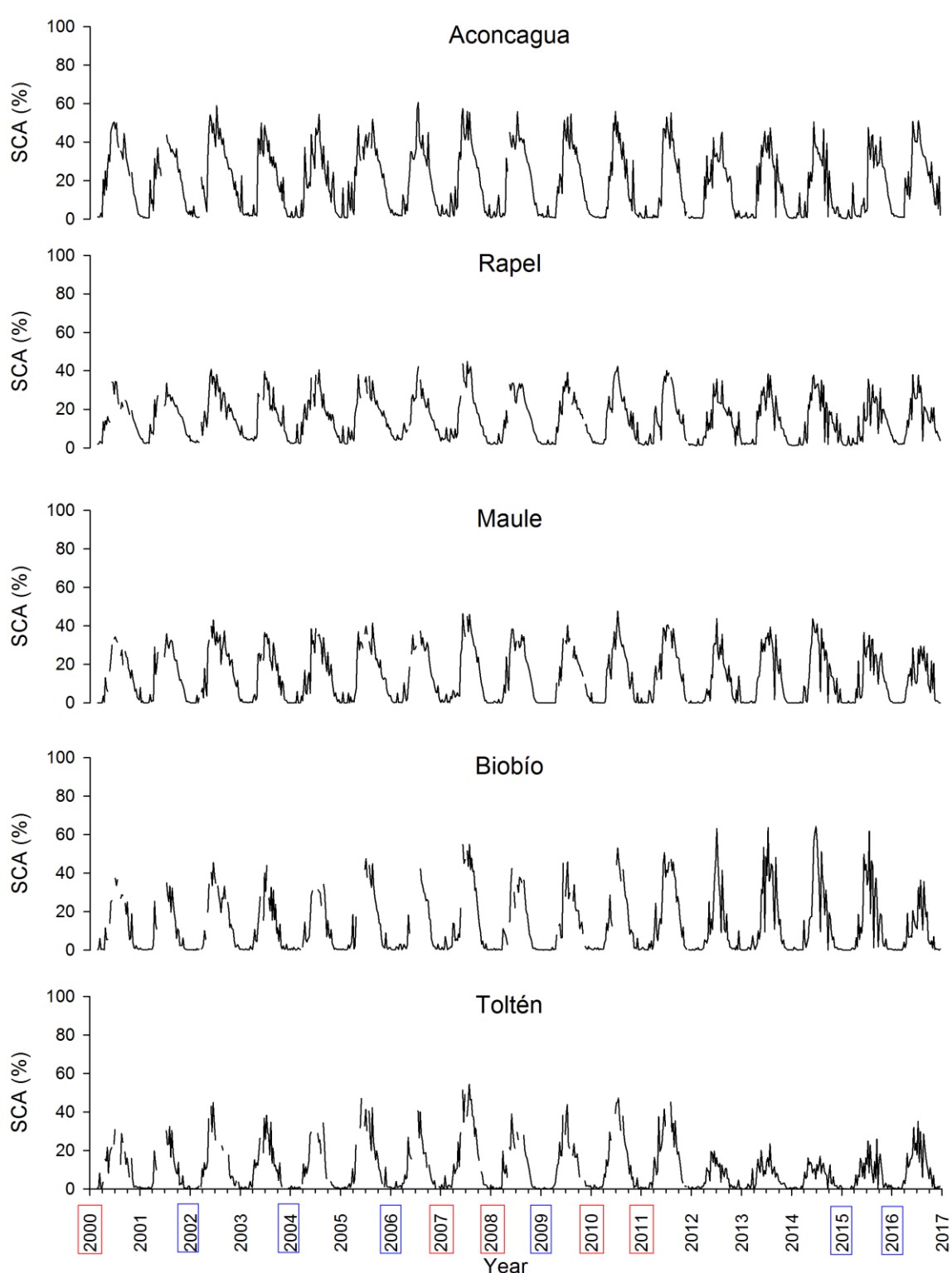

2    **Figure 4: SCD for the 2000–2016 period at Aconcagua, Rapel, Maule, Biobío and Toltén watersheds. Red boxes indicate**

3    **Niña years, blue ones Niño years.**

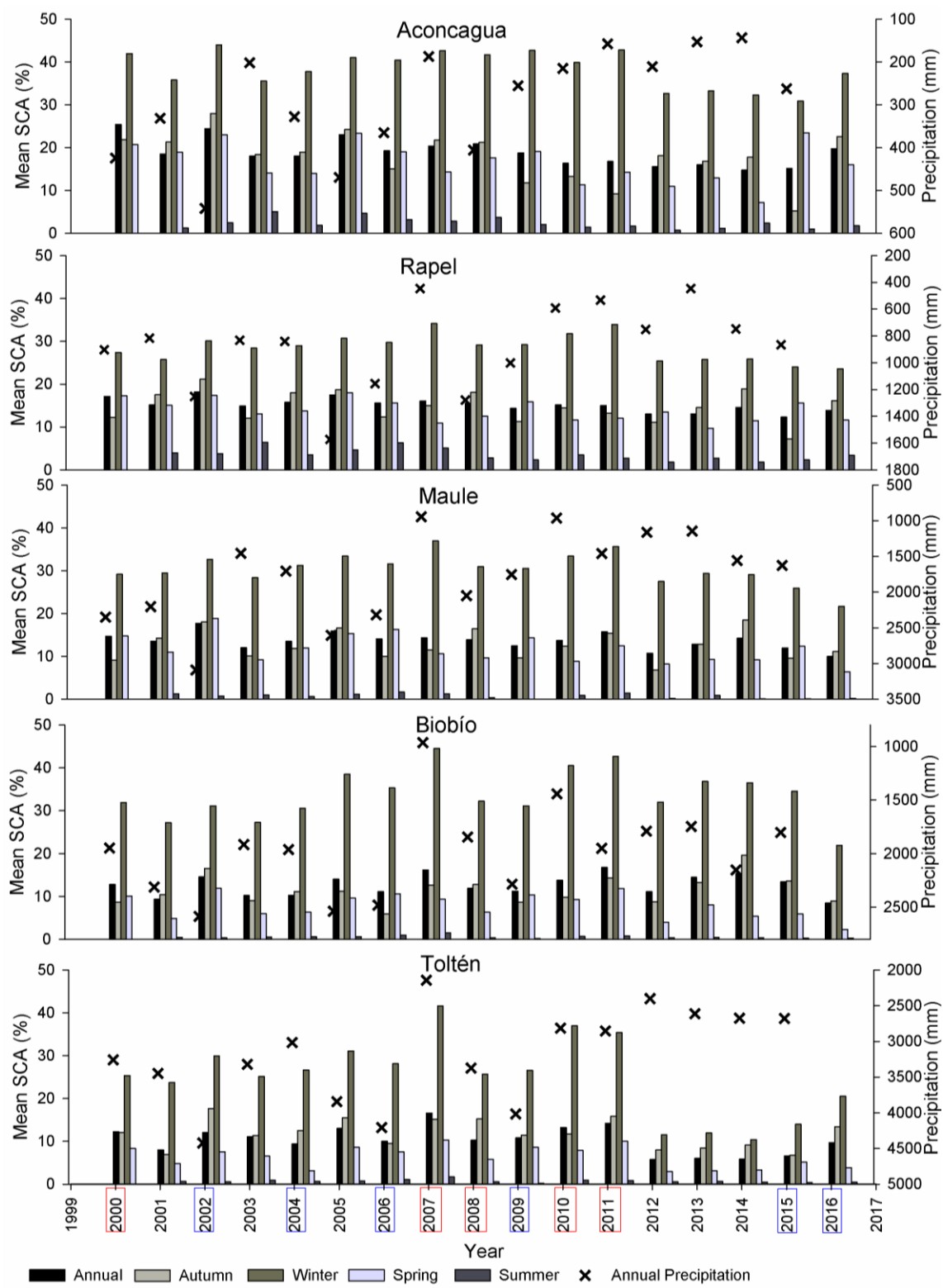

Figure 5: Mean seasonal and annual SCA (%) in Aconcagua, Rapel, Maule, Biobío and Toltén watersheds during 2000–

2016. Red boxes indicate Niña years, blue ones Niño years.

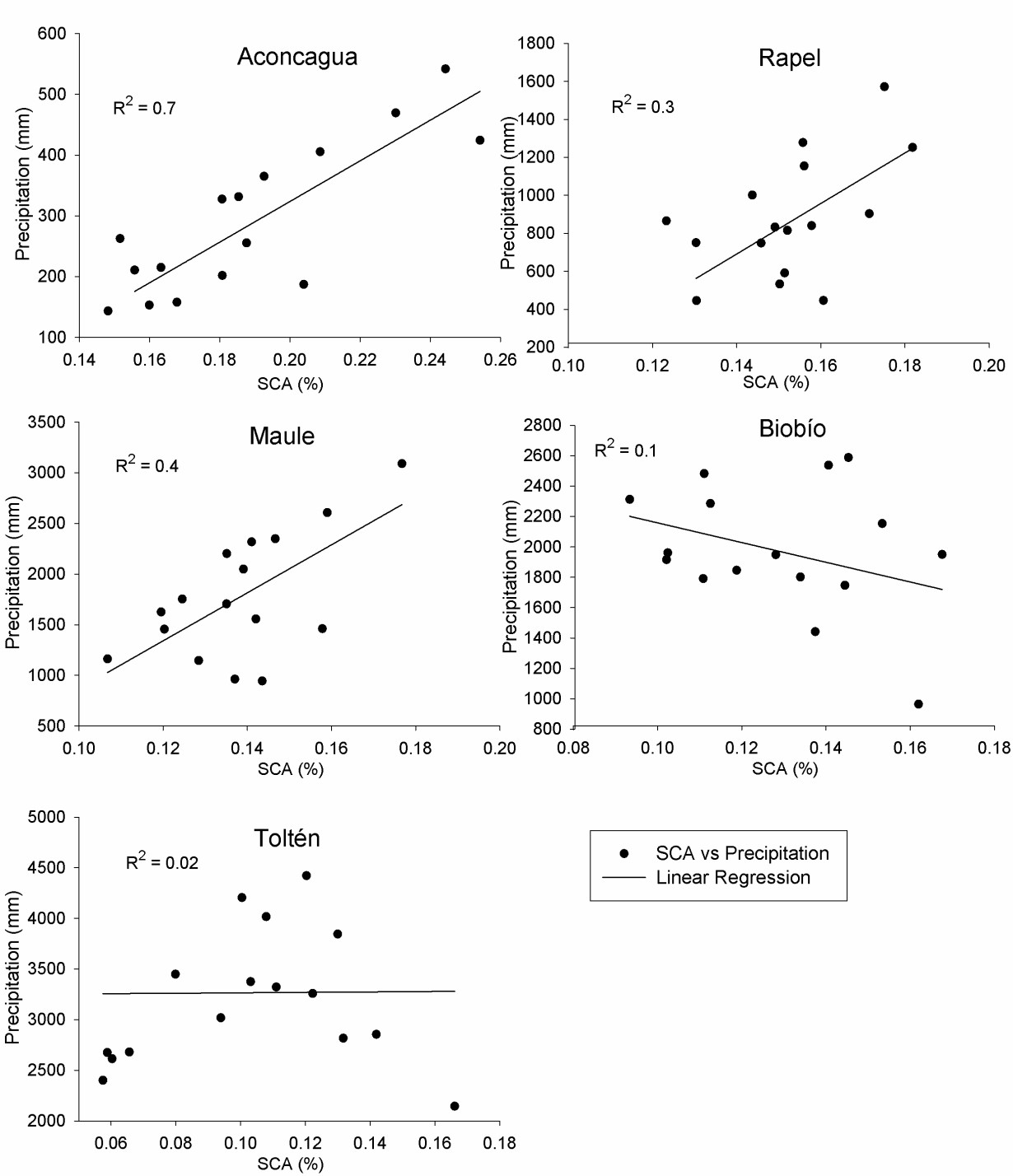

**Figure 6: Relation between % of mean annual SCA and annual precipitation at Aconcagua, Rapel, Maule, Biobío and Toltén watersheds.**

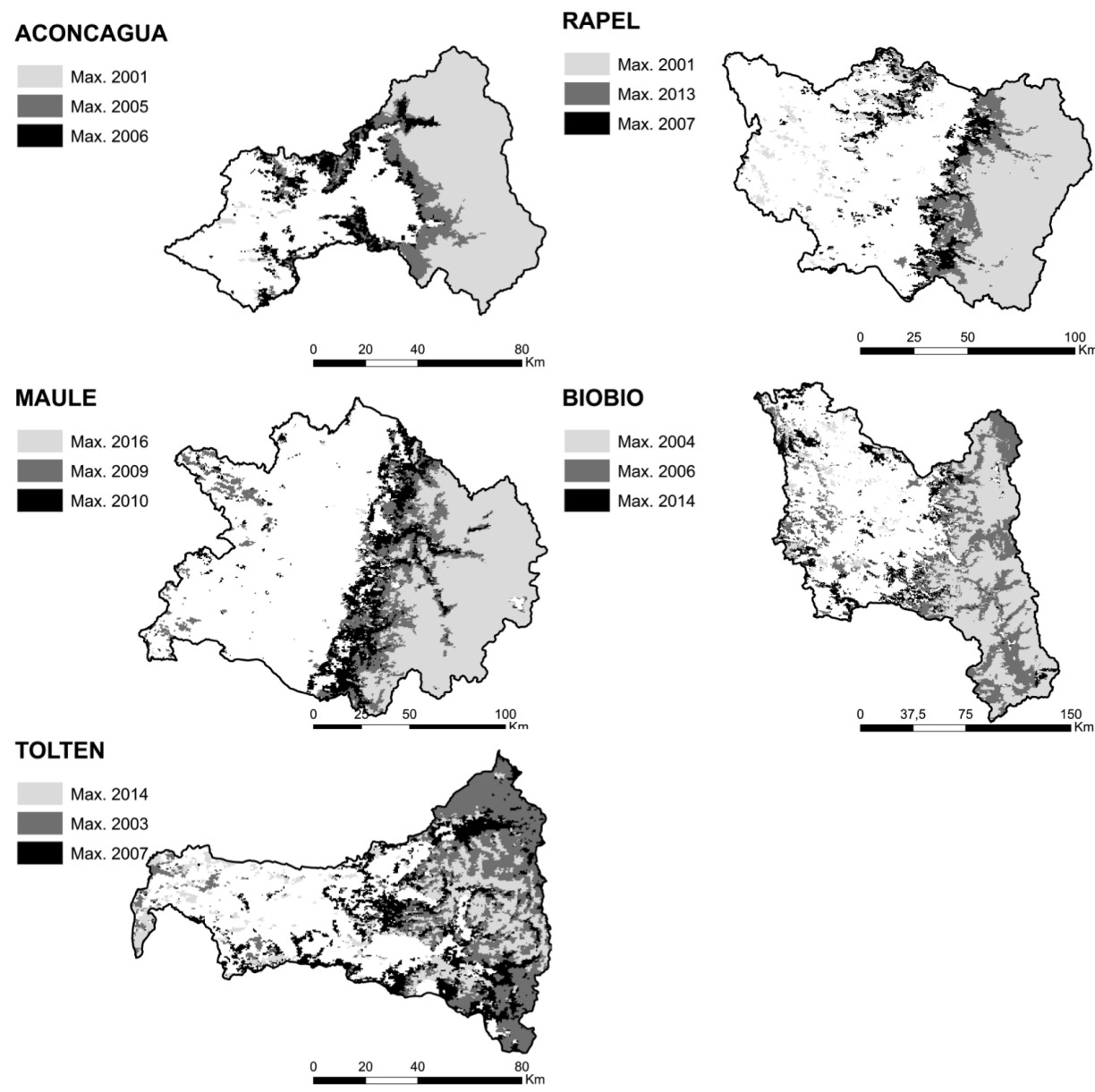

**Figure 7: Spatial representation of SCA for the period 2000 - 2016 in Aconcagua, Rapel, Maule, Biobío and Toltén watersheds. Black indicates the year with maximum SCA, dark grey the year with average SCA and grey the year with minimum SCA.**

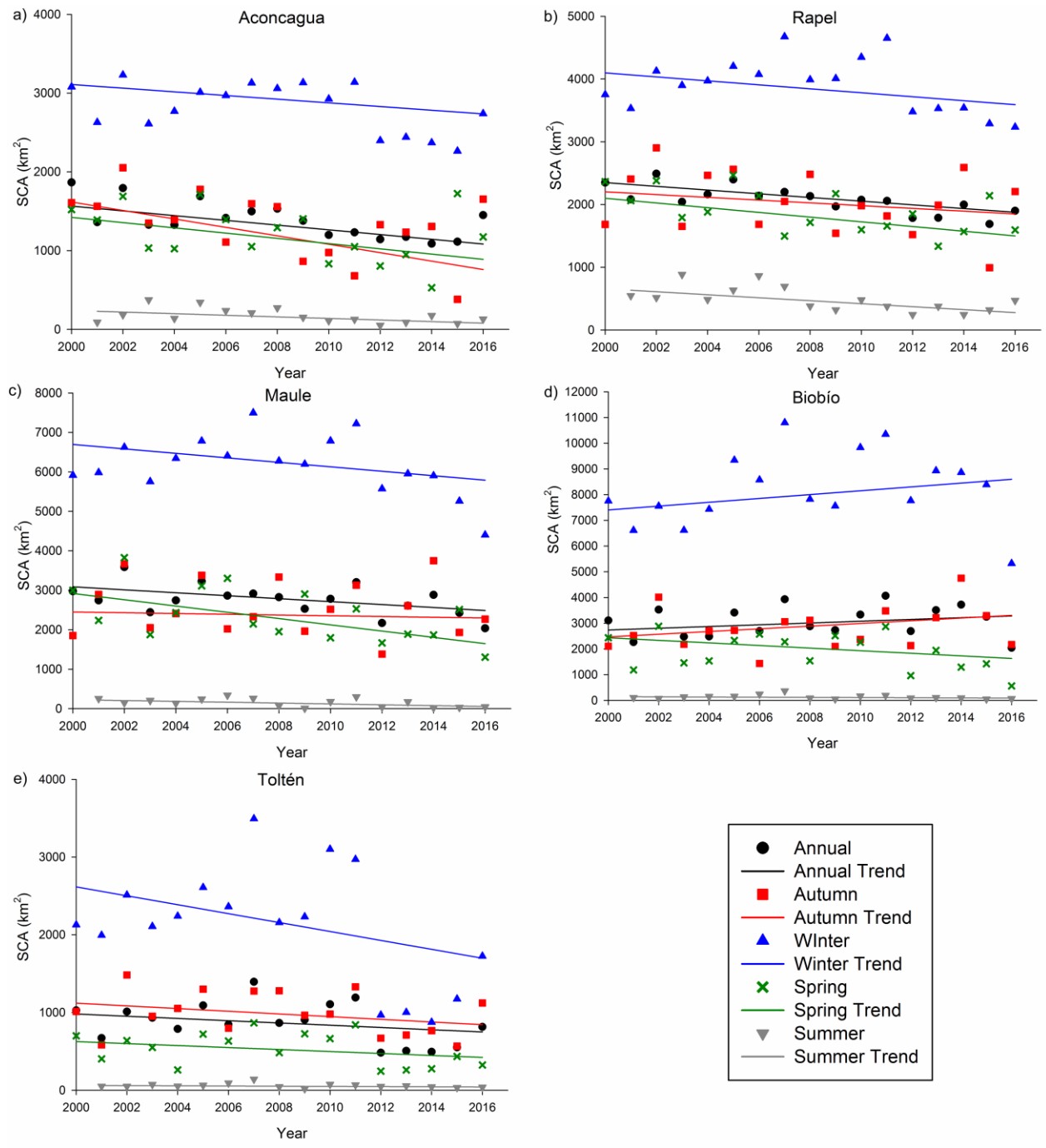

2 **Figure 8: Annual and seasonal trend of SCA at Aconcagua, Rapel, Maule, Biobío and Toltén watersheds.**