# Peer review of "Snow cover dynamics in Andean watersheds of Chile (32.0-39.5°S) during the years 2000 - 2016"

_Hydrology and Earth System Sciences, 2016_

## Referee Comment (RC1) · Anonymous Referee #1 · 20 Oct 2016

General comments

The main objective of this study is to evaluate the suitability of MODIS 8-day snow cover product (MOD10A2) to estimate snow cover area and duration in five watersheds in Chilean Andes. The MODIS dataset was validated against observed snow depth at a few ground base site observations and snow courses data from the period 2000–2013. The changes in snow cover area and duration were analysed by trend analysis. Results show that the overall Accuracy of MODIS10A2 ranges from 81 to 98%. Authors conclude that the evaluation of trends does not indicate significant changes in snow cover area and duration.

Overall I agree with the authors that the effects of snow accumulation and melt changes

in Chilean Andes can have significant impacts on allocation and using water resources. So the evaluation of spatial and temporal patterns of snow cover is important and interesting. On the other hand, however, I wonder to what extent the present manuscript contributes to some novel significant scientific contribution. The main research question here is not clearly formulated. The evaluation of suitability is rather vague and it is not clear in which respect it is novel. Why do the authors expect that the accuracy of MODIS snow cover products will be different in their region (Southern hemisphere)? There are plenty of studies performed in the past evaluating the accuracy of MODIS products, which show good agreement of satellite images with ground stations or other remote sensing data. The new validation presented in the manuscript is not very robust and not adding much value to these existing results. The number and elevation of ground observations is not representative (in terms of both- spatial and temporal resolution), so the conclusions drawn from these findings are rather over-interpreted and/or not justified well by presented results. The assessment of snow changes is an interesting topic, however the time period used here is rather short, so the interpretation is very challenging and not very robust. Some more deep analysis of the factors that control the variability of snow cover area and duration would be very interesting here. Overall the result section is very short (only 3 figures), so the significance of the results is not clearly demonstrated. The discussion is not linking the findings with existing literature, so the added value of results is not clear.

I would not recommend to publish this paper in the current form and suggest a very important revision of the paper.

Specific comments

1) Introduction, p.2, l.22-30: Please consider to revise this part. It is not clear how this is related to the main objective of the study. Maybe reduction of clouds in MODIS images is an important factor, which is worth to mention.

2) P.4: What is the meaning of snowline?

3) P.5, l.1: snowline around 850 is already presented before. Please consider to reduce duplication of information.

4) Is Fig.3 needed? I would suggest to use terminology used in previous literature (e.g. overall accuracy index, over-, underestimation errors, etc.)

5) P.7: How is SCD estimated?

6) Section 3.2: Please be more detailed what do we see in Fig.4, 5?

7) Why is the analysed period 2000-2013 and not 2015/6? Are the trends the same if the period would be longer? Is the 8-day product accurate enough to estimate the snow cover duration and its changes? Is the duration similar as would be estimated from daily product?

———————————————————

---

## Referee Comment (RC2) · Anonymous Referee #2 · 31 Oct 2016

The authors present an extensive analysis on the use of snow cover information derived from MODIS imagery for five different watersheds in Chile. The paper evaluates the remote sensing product against detailed field measurements, and analyzes (inter)annual variability and change. Although the paper gives an interesting view on snow cover behavior over several basins in the country, I have the following general observations: 1) Elaborate field measurements were used to evaluate the performance of the MOD10A2 product over two watersheds, which is the most significant contribution of this paper. Nevertheless, the DGA has multiple data points with historical snow depth and snow water equivalent. Why were these not used in the validation exercise? Given that this data complements the current analysis, and can be freely obtained from

the DGA, the authors are suggested to take these data into account. 2) The analysis of the Snow Cover Dynamic is rather limited, and should be further expanded to take into account previous trend analysis already published for Chile. Although most literature has focused on temperature and precipitation trends, this should provide relevant complementary information when comparing with snow dynamics. 3) The discussion on the influence of Niña or Niño years is also underdeveloped and more than observing apparent trends, the discussion should go more into depth why these ENSO years are markedly different with respect to snow cover. La Niña years are generally considered years with lower rainfall amounts in central Chile, which makes the observation that La Niña years provide more snow cover rather surprising. It therefore needs a more in depth analysis, why this apparent contradiction occurs. The timing of ENSO should also be taken into account, as ENSO is often strongest over the dry summer season in Chile, generating a delayed impact in the next winter season. The same is the case for the el Niño years, where you would expect significantly more snow volumes and higher snow cover as well, due to higher rainfall volumes. Specifically regarding snow cover, the influence of the Pacific Decadal Oscillation (PDO) has been shown to be a relevant factor, but this has not been addressed in the paper. 4) The selection of drought years was also done without providing sufficient detail in annual variability, especially considering that Chile has suffered a multi-annual drought in recent years (2010-2015). It is suggested to provide climatic information regarding the different years of study, by incorporating average rainfall amounts (as well as some additional meteorological indicators, such as minimum temperature) in the graphs or in a separate table. This should allow discussing more in detail what the different driver are for snow cover in those selected watersheds. 5) The paper has several English grammar errors and many typos, which should be corrected. It is strongly recommended that the paper is revised by an English native speaker for improved readability.

My additional comments are: P1-L14: typo P1-L15: not a sentence P1-L17: where is used instead of were in multiple occasions in the paper. Please correct them in all cases. P1-L21: this should read 'to' protect. P1-L26: where instead of were P3-L15:

comma too much P4-L26: truism instead of tourism. P5-L30: these instead of this

P5-L30: use instead of used

P6-L13: 'done' is not very specific here. What was done?

P6-L15: 'SCA and clouds in was quantified. . .' – please reformulate sentence.

P7-L4: whit instead of with P7-L5: I think this should be 'MODIS images' instead of 'MODIS' P7-L24: This sentence is redundant, as you already mention that the Bio Bio watershed is the only showing a trend. P7-L25: In spring should be used instead of 'At spring' P7-L27: In summer should be used instead of 'At summer' P7-L28: this should be show instead of shows P8-L1: where instead of were P8-L2: This should tread 'these two periods' instead of 'this two periods' P8-L19: 'in all watersheds' instead of 'an all watersheds' P9 – L4: analyzed instead of analyze P9 – L12: 'where compared' should be 'were compared' Fig 2: 'Rivers' instead of 'Rvers' Fig 4: ENSO years should be clearly marked on the figures Fig 5: ENSO years should be clearly marked on the figures

---

## Author Comment (AC1) · 7 Dec 2016

Regarding the scientific contribution of the paper, in Chile there are no studies which evaluated Snowcover dynamic (SCD) in watersheds through a latitudinal gradient (32° - 39.5°), which implies different climatic conditions. There is also no objective evidence of the influence of recent climatic variability in the extension of snow cover area in a wide special scale (Central- South Chile). I agree that the scientific question of the paper is not clear; we want to have a tool (MOD10A2) that allows us to analyze SCD in Chile, and get objevtive evidence of SCA changes. In that context there are very few meteorological stations and snow routes that measure snow, for example at the Biobio watershed there are none, so MODIS could be a good tool, as it is freely available.

[Figure]

On an improved version of the paper we can explain the research question better. The Andes mountain range (on the Chilean side) has a great oceanic effect due to the short distance between the coastal line and the mountains, unlike the northern hemisphere that has a great continental influence. This makes the isotherm 0 is higher than in the northern hemisphere; in addition the topographic features produce a great orographic effect, strongly affecting the regime of precipitation and temperatures. This causes small-scale spatial variations in weather, which is sometimes difficult to identify in satellite imagery. All the previously mention make it important to do a validation of MOD10A2, before using it for other analysis. For the research the daily product of MODIS was also tested with no good results, precisely because of all explained above. Regarding the number and elevation of ground observations, in a revised version of the manuscript we will incorporate more existing measurements, which will allow improving the conclusions. Concerning the time period that was used, they were the data we had available at that time. In case of submitting a revised manuscript we can easily incorporate the last 3 years (2014 - 2016). The referee suggest some more deep analysis of the factors that control the variability of snow cover area and duration , in this sense in a revised version it is possible to incorporate the spatial and not only the temporal analysis of the snow cover, and also an analysis of the relation between precipitation, temperature and snow cover. Regarding discussion this can be improved as suggested by the referee.

———————————————

---

## Author Comment (AC2) · 7 Dec 2016

In the next lines we give response to each of the observations done by the referee. 1) Regarding the number and elevation of ground observations, in a revised version of the manuscript we will incorporate more existing measurements, which will allow improving the conclusions. We have only used measurements done by ourselves, DGA data will be incorporated for validation of MOD10A2. 2) We agree with the suggestion, so in a revised version we will compare and discuses results with respect to previously published analysis in Chile (trend in precipitation and temperature). 3) The paper only analyzes snow coverage and not depth, which implicates that more SCA is not necessary more water stored as snow. This is something we can discuss more in detail.

From the existing data of the DGA, we will evaluate if it is possible to carry out an analysis changes in snow depth for the same period that we have analyze changes in SCA. The analysis will be done only in the watersheds with available data, depending on the length of existing time series. 4) Concerning the time period that was used, they were the data we had available at that time. In case of submitting a revised manuscript we can easily incorporate the last 3 years (2014 - 2016), which will allow approaching the period indicated by the reviewer. 5) English will be reviewed.

---

## Author Response (AR1)

**Response to comments**

**Anonymous Referee #1**

General Comments

The main objective of this study is to evaluate the suitability of MODIS 8-day snow cover product (MOD1 OA2) to estimate snow cover area and duration in five watersheds in Chilean Andes. The MODIS dataset was validated against observed snow depth at a few ground base site observations and snow courses data from the period 2000- 2013. The changes in snow cover area and duration were analyzed by trend analysis. Results show that the overall Accuracy of MODIS1 OA2 ranges from 81 to 98%. Authors conclude that the evaluation of trends does not indicate significant changes in snow cover area and duration.

Overall I agree with the authors that the effects of snow accumulation and melt changes in Chilean Andes can have significant impacts on allocation and using water resources.

So the evaluation of spatial and temporal patterns of snow cover is important and interesting. On the other hand, however, I wonder to what extent the present manuscript contributes to some novel significant scientific contribution.

*R: Regarding the scientific contribution of the paper, in Chile there are no studies which evaluated Snowcover dynamic (SCD) in watersheds through a latitudinal gradient (32º - 39.5º), which implies different climatic conditions. There is also no objective evidence of the influence of recent climatic variability in the extension of snow cover area in a wide special scale (Central- South Chile).*

The main research question here is not clearly formulated.

*R: We have formulated the research question again.*

The evaluation of suitability is rather vague and it is not clear in which respect it is novel.Why do the authors expect that the accuracy of MODIS snow cover products will be different in their region (Southern hemisphere)?

*R: The Andes mountain range (on the Chilean side) has a great oceanic effect due to the short distance between the coastal line and the mountains, unlike the northern hemisphere that has a great continental influence. This makes the isotherm 0 is higher than in the northern hemisphere; in addition the topographic features produce a great orographic effect, strongly affecting the regime of precipitation and temperatures. This causes small-scale spatial variations in weather, which is sometimes difficult to identify in satellite imagery. All the previously mention make it important to do a validation of MOD10A2, before using it for other analysis. For the research the daily product of MODIS was also tested with no good results, precisely because of all explained above.*

There are plenty of studies performed in the past evaluating the accuracy of MODIS products, which show good agreement of satellite images with ground stations or other remote sensing data. The new validation presented in the manuscript is not very robust and not adding much value to these existing results. The number and elevation of ground observations is not representative (in terms of both- spatial and temporal resolution), so the conclusions drawn from these findings are rather over-interpreted and/or not justified well by presented results.

*R: Regarding the number and elevation of ground observations, in a revised version of the manuscript we will incorporate more existing measurements, which will allow improving the conclusions*

The assessment of snow changes is an interesting topic, however the time period used here is rather short, so the interpretation is very challenging and not very robust. Some more deep analysis of the factors that control the variability of snow cover area and duration would be very interesting here.

*R: Concerning the time period that was used, they were the data we had available at that time. In case of submitting a revised manuscript we can easily incorporate the last 3 years (2014 - 2016).*

*The referee suggest some more deep analysis of the factors that control the variability of snow cover area and duration , in this sense we have incorporate the spatial and not only the temporal analysis of the snow cover, and also an analysis of the relation between precipitation and snow cover.*

Overall the result section is very short (only 3 figures), so the significance of the results is not clearly demonstrated.

*R: We have added some figures and explain more each of them.*

The discussion is not linking the findings with existing literature, so the added value of results is not clear.

*R: We have added at the discussion the linkage with existing literature.*

Specific comments

1) Introduction, p.2, 1.22-30: Please consider to revise this part. It is not clear how this is related to the main objective of the study. Maybe reduction of clouds in MODIS images is an important factor, which is worth to mention.

*R: Was revised and change*

2) P.4: What is the meaning of snowline?

*R: Mean altitude above which there is snow in winter, definition was added in the manuscript.*

3) P.5, 1.1: snowline around 850 is already presented before. Please consider to reduce duplication of information.

*R: Corrected*

4) Is Fig.3 needed? 1 would suggest to use terminology used in previous literature (e.g. overall accuracy index, over-, underestimation errors, etc.)

*R: Was change according the suggestion of the referee*

5) P.7: How is SCD estimated?

*R: Explained better in the text.*

6) Section 3.2: Please be more detailed what do we see in Fig.4, 5?

*R: We have explain in more detail both figures*

?) Why is the analyzed period 2000-2013 and not 2015/6? Are the trends the same if the period would be longer? Is the 8-day product accurate enough to estimate the snow cover duration and its changes? Is the duration similar as would be estimated from daily product?

R: We have incorporated the last 3 years (2014 - 2016). For the research the daily product of MODIS was also tested with no good results.

**Anonymous Referee #2**

The authors present an extensive analysis on the use of snow cover information derived from MODIS imagery for five different watersheds in Chile. The paper evaluates the remote sensing product against detailed field measurements, and analyzes (inter)annual variability and change. Although the paper gives an interesting view on snow cover behavior over several basins in the country, I have the following general observations:

1) Elaborate field measurements were used to evaluate the performance of the MOD10A2 product over two watersheds, which is the most significant contribution of this paper. Nevertheless, the DGA has multiple data points with historical snow depth and snow water equivalent. Why were these not used in the validation exercise? Given that this data complements the current analysis, and can be freely obtained from the DGA, the authors are suggested to take these data into account.

*R: Regarding the number and elevation of ground observations, we have incorporated more existing measurements from DGA for validation of MOD10A2.*

2) The analysis of the Snow Cover Dynamic is rather limited, and should be further expanded to take into account previous trend analysis already published for Chile. Although most literature has focused on temperature and precipitation trends, this should provide relevant complementary information when comparing with snow dynamics.

*R: We have added in discussion the comparison with previous studies.*

3) The discussion on the influence of Niña or Niño years is al so underdeveloped and more than observing apparent trends, the discussion should go more into depth why these ENSO years are markedly different with respect to snow cover. La Niña years are generally considered years with lower rainfall amounts in central Chile, which makes the observation that La Niña years provide more snow cover rather surprising. It therefore needs a more in depth analysis, why this apparent contradiction occurs. The timing of ENSO should also be taken into account, as ENSO is often strongest over the dry summer season in Chile, generating a delayed impact in the next winter season. The same is the case for the el Niño years, where you would expect significantly more snow volumes and higher snow cover as well, due to higher rainfall volumes. Specifically regarding snow cover, the influence of the Pacific Decadal Oscillation (PDO) has been shown to be a relevant factor, but this has not been addressed in the paper.

*R: We have discussed in more detail the influence of Niña and Niño years. Regarding PDO this has been no taken into account.*

4) The selection of drought years was also done without providing sufficient detail in annual variability, especially considering that Chile has suffered a multi-annual drought in recent years (2010-2015). It is suggested to provide climatic information regarding the different years of study, by incorporating average rainfall amounts (as well as some additional meteorological indicators, such as minimum temperature) in the graphs or in a separate table. This should allow discussing more in detail what the different driver are for snow cover in those selected watersheds.

*R: We have incorporated data about annual precipitation for the upper part of the different watersheds. We have not incorporate temperature as temperature records where very incomplete.*

5) The paper has several English grammar errors and many typos, which should be corrected. It is strongly recommended that the paper is revised by an English native speaker for improved readability.

*R: A person with high English knowledge has revised the paper.*

My additional comments are:

P1-L 14: typo

*R: Corrected*

P1-L 15: not a sentence

5  *R: Corrected*

P1-L 17: where is used instead of were in multiple occasions in the paper. Please correct them in all cases.

*R: Corrected*

P1-L21: this should read 'to' protect.

*R: Corrected*

10  P1-L26: where instead of were P3-L 15: comma too much

*R: Corrected*

P4-L26: truism instead of tourism. P5-L30: these instead of this

*R: Corrected*

P5-L30: use instead of used

15  *R: Corrected*

P6-L 13: 'done' is not very specific here. What was done?

*R: Corrected*

P6-L 15: 'SCA and clouds in was quantified .. .' - please reformulate sentence.

*R: Corrected*

20  P7-L4: whit instead of with P7-L5: I think this should be 'MODIS images' instead of 'MODIS' P7-L24: This sentence is redundant, as you already mention that the Bio Bio watershed is the only showing a trend.

*R: Corrected*

P7-L25: In spring should be used instead of 'At spring'

*R: Corrected*

25  P7-L27: In summer should be used instead of 'At summer'

*R: Corrected*

P7-L28: this should be show instead of shows

*R: Corrected*

P8-L 1: where instead of were P8-L2: This should tread 'these two periods' instead of 'this two periods'

30  *R: Corrected*

P8-L 19: 'in all watersheds' instead of 'an all watersheds'

*R: Corrected*

P9 - L4: analyzed instead of analyze

*R: Corrected*

P9 - L 12: 'where compared' should be 'were compared' Fig 2: 'Rivers' instead of 'Rvers' Fig 4: ENSO years should be clearly marked on the figures Fig 5: ENSO years should be clearly marked on the figures

*R: Corrected*

[revised manuscript text omitted]

**ACONCAGUA**
- Max. 2001
- Max. 2005
- Max. 2006

**RAPEL**
- Max. 2001
- Max. 2013
- Max. 2007

**MAULE**
- Max. 2016
- Max. 2009
- Max. 2010

**BIOBIO**
- Max. 2004
- Max. 2006
- Max. 2014

**TOLTEN**
- Max. 2014
- Max. 2003
- Max. 2007

Figure 6: Spatial representation of SCA for the period 2000 - 2016 in Aconcagua, Rapel, Maule, Biobío and Toltén watersheds. Black indicates the year with maximum SCA, dark grey the year with average SCA and gray the year with minimum SCA.

[Figure]

[Figure]

Figure 67: Annual and seasonal trend of SCA at Aconcagua, Rapel, Maule, Biobío and Toltén watersheds.

---

## Referee Report (RR1)

**Snow cover dynamics in Andean watersheds of Chile (32.0-39.5°S) during the years 2000-2013**

by Aljandra Stehr and Mauricio Aguayo

This study analyses the snow cover area (SCA) and snow cover dynamics (SCD) for Chile using MODIS 8-day snow cover data. The authors first conduct evaluation of MODIS 8-day data using in-situ data and then analyze trends in the last years (since 2000). Moreover, they look the correlation of SCA with precipitation data. In, general I agree with the authors that snow information is highly valuable for such mountain regions where snow is stored in winter months and contributes to discharge in warm season when water resources is important for agricultural and energy production purposes. MODIS snow cover product can therefore be highly valuable in such remote areas where observation is rare. Therefore, I support the idea to use RS snow cover data to better understand the hydrological behavior in alpine regions.

However, in my opinion the study lacks on scientific quality. The outcomes that are based on scientific results may therefore not be robust and valid. The presented figures also have less quality. Moreover, some sentences are contradictory to results and needs justification. The outcomes (e.g. evaluation of MODIS data, trend analysis, correlations, etc.) that should bring scientific novelty are not based on robust methods which makes them questionable at the end. Therefore, I would not suggest this study for publication in HESS. My conclusion is based mainly on following general comments:

1.  The authors evaluate MODIS 8-day snow cover data using in-situ data. MODIS 8-day snow cover data is the composite (maximum extent) of daily MODIS snow cover data (MOD10A1). If the authors compare average in-situ snow depth data for these 8 days with maximum snow cover extent in the same 8 days period, then the comparison is inappropriate as only one day of maximum snow cover extent can have high impact on 8-day snow cover data whereas this is not the case with in-situ data. Moreover, the authors state that they did not achieve good results for evaluation of daily snow cover data against in-situ, but the 8-day snow cover data is the product of daily snow cover data. Therefore, in my opinion the evaluation is not comprehensive. It would be more interesting to use daily snow cover data for evaluation to understand the impact of local climate (e.g. due to close distance to ocean, steep slopes) on snow cover behavior and why this can be challenging for snow observation using remote sensing.

2.  The authors use only 13 (16) years of data for trend analysis. In my opinion, this is too less time series to carry out trend analysis and to make meaningful and significant conclusions on snow cover variability in this region. Moreover, it is not clear which time frame is used at the end for trend analysis as there are phrases with data usage on 2000-2013 vs. 2000-2016 (in the abstract section).

3.  The presented quality of research is poor. The correlation analysis between SCA and precipitation also does not add any additional value to the research. It is known that precipitation should to some extent control the snow cover and therefore some correlation is expected (in most cases positive correlation). The relationship between SCA and precipitation as illustrated in Fig. 5 (for Biobio and Tolten watersheds) should not lead to conclusions that there is a negative or no correlation as the points are scattered over all region with no clear dependency.

4.  The authors state in the text that they used also temperature data for the analysis (page 3, lines 33-34) but this was not the case.

5. Snowline definition is based on different sources for different watersheds and these studies are also from 1990s. High variation between snowlines (2100 m vs. 750 m) for different watersheds are questionable. It is difficult for international community to follow on which data are the values for snowline are based since the references are also in other language (Spanish).

Specific comments:

1. Figure 2 is too detailed and confusing. There is much more information than needed. For instance differentiation of each snow routines or observations is not necessary. Also, the duplication of figures as in Fig. 2b and 2d is not necessary.
2. Figure 3 shows some questionable results. There are several "jumps" in SCA after 2012/2013 where snow cover almost disappears for several days (or only one observation period – 8 day) and reaches its previous value only after few days. Such evidence is not visible in the time series before 2012/2013. The authors should check whether this is not due to systematic errors.
3. Figure 4 does not add any additional scientific value to this research.

Due to my general comments that leads to rejection, I did not add any further specific comments although there are some more where presentation art does not fit the standards of HESS.

---

## Author Response (AR2)

**Response to comments**

**Anonymous Referee #1**

General comments

I appreciate the revisions made by authors. However, I feel that there is still some revision needed (in order to make the contribution of the manuscript more significant). I would suggest to consider following points for a revision:

1) I like the added motivation and formulations on why and how is the Andes region different to the other parts of the world. These specific conditions need to be one of the main selling points of the manuscript. I would therefore recommend to further strengthen this message in the introduction and Discussion sections. The Introduction in its current form (e.g the part between p.7, l.30 – p.8, l.20 in the document with marked changes) is quit randomly organised (in terms of the main objectives of the paper). A more focused literature overview would be needed here. I would suggest to add references related to the changes (inter/intra annual variability) in snow cover area, snowline elevation, duration , etc. derived from MODIS images in other parts of the world. This will then allow to compare your results with other regions and demonstrate more clearly the value of your findings (differences and similarities obtained in such diverse climate region of Andes). (please look e.g. to literature in recent paper of Krajci et al., 2016, Variability of snow line elevation, snow cover area and depletion in the main Slovak basins in winters 2001-2014.).

*R: We added new references, and the introduction was rewritten.*

2) Please consider to reformulate the main objectives of the paper. "Evaluate the suitability of MODIS as a tool" is probably not a very scientific research question. I would suggest to highlight the aim to analyse and evaluate the changes in spatial and temporal snow cover characteristics (you have selected) over the Andes and also to attribute these changes (if possible) to some climate characteristics (if possible).

*R: The main objective was reformulated.*

3) I really appreciate the extension of the period used for analysis (to 2016). Please make consistent this change across the entire manuscript (use just one period, 2000-2016, in the title, abstract, data, results).

*R: The paper was checked and we have changed in the paper.*

4) Discuss the findings more closely with respect to the literature related to the inter- and intra-annual variability of snow cover characteristics in other parts of the world (including factors that affect this change).

*R: the discussion was extended considering the referee suggestion.*

Specific comments

1) MODIS version. Please add which version of MODIS dataset is used (v005 or v 006?).

*R: We used v005, it was added on the text.*

2) Please consider to add the equations used for validation (estimating overall accuracy, etc.) to the Methods section (E.g. by adding a subsection on validation)

*R: A figure explaining was incorporated to explain validation indexes.*

3) Please be more specific how is the mean annual precipitation (over the basins) estimated?

*R: We added a more specific explanation on how mean annual precipitation was estimated for the part of the watershed cover by snow .*

4) Table 1 caption. Add the period in which the snow courses were repeated.

*R: Was added.*

5) Fig.3 and Fig. 4 for Toltén basin and period 2012-2016 are not consistent. Fig. 3 indicates even larger snow cover area in these years, but the mean is lower. Why? Please check and add some discussion.

*R: Figure 3 was wrong, it was corrected. The Graph at Toltén was the as Biobío.*

6) Fig. 7 is difficult to read. Perhaps, it will be more clear if the results will be organized according the basins (a panel for each basin and seasonal changes within this panel).

*R: Results were reorganized.*

7) Please check the English (if this will not be a part of final proof-reading).

*R: English was check by English – Spanish translator.*

**Anonymous Referee #2**

General comments

1) The authors evaluate MODIS 8-day snow cover data using in-situ data. MODIS 8-day snow cover data is the composite (maximum extent) of daily MODIS snow cover data (MOD10A1). If the authors compare average in-situ snow depth data for these 8 days with maximum snow cover extent in the same 8 days period, then the comparison is inappropriate as only one day of maximum snow cover extent can have high impact on 8-day snow cover data whereas this is not the case with in-situ data. Moreover, the authors state that they did not achieve good results for evaluation of daily snow cover data against in-situ, but the 8-day snow cover data is the product of daily snow cover data. Therefore, in my opinion the evaluation is not comprehensive. It would be more interesting to use daily snow cover data for evaluation to understand the impact of local climate (e.g. due to close distance to ocean, steep slopes) on snow cover behavior and why this can be challenging for snow observation using remote sensing.

*R: We disagree with the comment done be the referee, as MOD10A2 is a composite image, it represents the maximum extent of SCA for the 8 days period, we have check this with in situ data.*

2) The authors use only 13 (16) years of data for trend analysis. In my opinion, this is too less time series to carry out trend analysis and to make meaningful and significant conclusions on snow cover variability in this region. Moreover, it is not clear which time frame is used at the end for trend analysis as there are phrases with data usage on 2000-2013 vs. 2000-2016 (in the abstract section).

*R: We used 17 year for trend analysis, the time frame is 2000 – 2016, and it was corrected in the manuscript. Unfortunately in Chile data availability on snow is very scarce and we do not have more availability of data. Studies on SCA trend in other parts of the world are done with this amount of data ore lesser. We think that a good start to for further studies.*

3) The presented quality of research is poor. The correlation analysis between SCA and precipitation also does not add any additional value to the research. It is known that precipitation should to some extent control the snow cover and therefore some correlation is expected (in most cases positive correlation). The relationship between SCA and precipitation as illustrated in Fig. 5 (for Biobio and Tolten watersheds) should not lead to conclusions that there is a negative or no correlation as the points are scattered over all region with no clear dependency.

*R: We agree that we cannot have a final conclusion about negative correlation of SCA with precipitation, the idea was to use also temperature, but there are not good quality measurements of temperature in the higher parts of the watershed.*

4) The authors state in the text that they used also temperature data for the analysis (page 3, lines 33-34) but this was not the case.

*R: That was a mistake in the text. We wanted to use temperature data but it was not possible.*

5) Snowline definition is based on different sources for different watersheds and these studies are also from 1990s. High variation between snowlines (2100 m vs. 750 m) for different watersheds are questionable. It is difficult for international community to follow on which data are the values for snowline are based since the references are also in other language (Spanish).

*R: There is no more recent studies and valid references about snowline in Chile that are in English. The variation between the snowlines high is due to climate characteristic of each watershed.*

Specific comments

1) Figure 2 is too detailed and confusing. There is much more information than needed. For instance differentiation of each snow routines or observations is not necessary. Also, the duplication of figures as in Fig. 2b and 2d is not necessary.

2) Figure 3 shows some questionable results. There are several "jumps" in SCA after 2012/2013 where snow cover almost disappears for several days (or only one observation period – 8 day) and reaches its previous value only after few days. Such evidence is not visible in the time series before 2012/2013. The authors should check whether this is not due to systematic errors.

*R: We have checked the data and they are ok.*

3. Figure 4 does not add any additional scientific value to this research.

[revised manuscript text omitted]

$$\begin{array}{c|cccccc}
k,k & A & B & C & \cdots & q & \Sigma \\
\hline
A & n_{AA} & n_{AB} & n_{AC} & \cdots & n_{Aq} & n_{A+} \\
B & n_{BA} & n_{BB} & n_{BC} & \cdots & n_{Bq} & n_{B+} \\
C & n_{CA} & n_{CB} & n_{CC} & \cdots & n_{Cq} & n_{C+} \\
\vdots & \vdots & \vdots & \vdots & \cdots & \vdots & \vdots \\
q & n_{qA} & n_{qB} & n_{qC} & \cdots & n_{qq} & n_{q+} \\
\Sigma & n_{+A} & n_{+B} & n_{+C} & \cdots & n_{+q} & n
\end{array}$$

$$\text{Percentage correct} = \frac{\sum_{k=1}^{q} n_{kk}}{n} \times 100$$

$$\text{User's accuracy} = \frac{n_{kk}}{n_{k+}} \times 100$$

$$\text{Producer's accuracy} = \frac{n_{kk}}{n_{+k}} \times 100$$

2  **Figure 3: Confusion matrix and percentage correct (overall accuracy), user accuracy and producer accuracy.**

[Figure]

[Figure]

Figure 3̶4: SCD for the ̶ ̶p̶e̶r̶i̶o̶d̶ 2000̶ ̶ ̶2̶0̶1̶3̶ 2016 period at Aconcagua, Rapel, Maule, Biobío and Toltén 6̶watersheds. Red

boxes indicate Niña years, blue ones Niño years.

[Figure]

Figure 45: Mean seasonal and annual SCA (%) in Aconcagua, Rapel, Maule, Biobío and Toltén watersheds during 2000–20132016. Red boxes indicate Niña years, blue ones Niño years.

[Figure]

2 **Figure 5̶6: Relation between % of mean annual SCA and annual precipitation at Aconcagua, Rapel, Maule, Biobío and**

3 **Toltén watersheds.**

[Figure]

Figure 67: Spatial representation of SCA for the period 2000 - 2016 in Aconcagua, Rapel, Maule, Biobío and Toltén
watersheds. Black indicates the year with maximum SCA, dark grey the year with average SCA and grey the year
with minimum SCA.

[Figure]

[Figure]

2 **Figure 78: Annual and seasonal trend of SCA at Aconcagua, Rapel, Maule, Biobío and Toltén watersheds.**